# Demonstrating a Filter-Free Wavelength Sensor with Double-Well Structure and Its Application

**DOI:** 10.3390/bios12111033

**Published:** 2022-11-17

**Authors:** Yong-Joon Choi, Kakeru Nakano, Tomoya Ide, Tsugumi Sakae, Ryosuke Ichikawa, Takeshi Hizawa, Daisuke Akai, Kazuhiro Takahashi, Toshihiko Noda, Kazuaki Sawada

**Affiliations:** 1Department of Electrical and Electronic Information Engineering, Toyohashi University of Technology, Toyohashi 441-8580, Japan; 2Office for Technical Support Services, Toyohashi University of Technology, Toyohashi 441-8580, Japan; 3Electronics Inspired-Interdisciplinary Research Institute (EIIRIS), Toyohashi University of Technology, Toyohashi 441-8580, Japan

**Keywords:** filter-free, wavelength detection, double-well, photogate type sensor, multiplex, fluorescence detection, fluorescent reagent, centroid wavelength

## Abstract

This study proposed a filter-free wavelength sensor with a double-well structure for detecting fluorescence without an optical filter. The impurity concentration was optimized and simulated to form a double-well-structured sensor, of which the result was consistent with the fabricated sensor. Furthermore, we proposed a novel wavelength detection method using the current ratio based on the silicon absorption coefficient. The results showed that the proposed method successfully detected single wavelengths in the 460–800 nm range. Additionally, we confirmed that quantification was possible using the current ratio of the sensor for a relatively wide band wavelength, such as fluorescence. Finally, the fluorescence that was emitted from the reagents ALEXA488, 594, and 680 was successfully identified and quantified. The proposed sensor can detect wavelengths without optical filters, which can be used in various applications in the biofield, such as POCT as a miniaturized wavelength detection sensor.

## 1. Introduction

Optical detection techniques can predict, diagnose, and analyze diseases as a compact system by detecting the optical properties of a substance, such as absorption [1], fluorescence [2], and luminescence [3], and hence, they are widely used as measurement and analysis equipment in various fields, such as in medicine [4], the environment [5], in chemicals [6], in food [7], in biology [8,9], and in the military [10]. In general, silicon-based photodiodes detect light in various applications, but high-performance photodetectors are required to obtain accurate and fast information. Therefore, by integrating silicon and specific materials, photodetectors which have a broad detection wavelength range from ultraviolet to infrared rays, a high quantum efficiency, and fast response characteristics have been reported [11,12]. Among these, the fluorescence detection method is the most helpful one because it contains a large amount of information, and it is easy to handle. Detection methods such as fluorescence intensity (FI) [13], fluorescence resonance energy transfer (FRET) [14], fluorescence polarization (FP) [15], and time-resolved fluorescence (TRF) [16] are applied using fluorescence with high selectivity and sensitivity to the detection target.

The fluorescence wavelength is selectively detected using a monochromator or an optical filter. These components were applied to a spectrofluorometer, which detects a specific wavelength by dispersing the light that is emitted from a sample using a diffraction grating or prism, and a fluorescence microscope, which can detect fluorescence with high sensitivity using an optical filter. Because simultaneous detection of multiple wavelengths is possible in a relatively wide band, it is advantageous for quantitative and qualitative analyses [17,18]. However, because the wavelength is detected by dispersing light at one point, it becomes challenging to image the sample. In addition, it is possible to selectively detect a relatively large area of fluorescence that is emitted from a sample [19,20]. However, since the detected fluorescence wavelength depends on the optical filter, various wavelengths cannot be detected simultaneously. Although these fluorescence detection devices have high sensitivity and selectivity, they are expensive and bulky owing to the integration of various optical filters and components, which makes applying them to point-of-care testing (POCT) a challenge as this requires it to be portable [21,22]. Therefore, various studies have been reported to realize the miniaturization, low cost, and high performance of the fluorescence detection systems.

On-chip fluorescence detection devices have been reported to integrate the interference or absorption filters into CMOS image sensors with high selectivity and sensitivity. Owing to these advantages, Ohta et al. developed an in vivo fluorescence detection device and successfully detected in vivo images of rats [23]. Additionally, a hybrid filter, where the absorption and interference filters are integrated devices, is reported, as shown in Figure 1a. Because it is possible to detect multiple wavelengths simultaneously, their applications in biofields such as resonance energy transfer (FRET) and multiplex fluorescence imaging analysis are expected [24]. However, because the optical filter is integrated into the CMOS image sensor, the detection of different wavelength changes in the fluorescent reagent is challenging.

Figure 1b shows a method of multi-wavelength analysis using a single pixel and a CMOS buried quad p-n junction photodiode structure [25,26]. Because the light wavelength has a different absorption depth depending on the silicon absorption coefficient, it is possible to separate the wavelength by measuring the current that is generated at each p-n junction. Furthermore, because the structure of such a buried multi-p-n junction can detect a wavelength in a single pixel, it has a higher fill factor than a CMOS image sensor with an integrated RGB filter does. Therefore, it provides a high-resolution fluorescence image in the biofield. However, the wavelength and band numbers were fixed according to the buried p-n junction numbers and their depth.

Previously, we reported a filter-free fluorescence sensor with a photogate structure to detect the light intensity of multiple wavelengths without using optical components, even when the excitation and fluorescence wavelengths are changed, as shown in Figure 1c [27,28]. The sensor with a single-well structure on an n-type silicon substrate adjusts the potential depth *W* multiple times by controlling the photogate (PG) voltage, and it detects only electrons that move toward the surface side from the adjacent electrodes *I*_PG_. However, because the electrons cannot be detected toward the substrate at depth *W*, the light reception sensitivity may decrease. Moreover, an error occurred in the measured value according to the change in the full width at half maximum (FWHM) of the incident light. Because the wavelength information of the excitation light and fluorescence was required to obtain the fluorescence intensity, it was impossible to detect the intensity of the unknown fluorescence. In addition, it was necessary to separate the independent wells to configure the peripheral circuit.

This study proposes an improved filter-free wavelength sensor with a double-well structure that can detect unknown wavelengths and integrate the peripheral circuits. The silicon-based impurity concentration conditions were simulated and evaluated to fabricate a filter-free wavelength sensor with a double-well structure. We report the experimental results of the wavelength dependence, light intensity dependence, and FWHM dependence of the fabricated sensor. Lastly, we report the measurement results for three fluorescent reagents with different wavelengths as an application experiment using the proposed sensor.

## 2. Design and Principle

Figure 1c shows a previously reported filter-free fluorescence sensor on n-type silicon substrate. A p-well layer was formed on the silicon substrate and an *n^+^* diffusion layer was arranged to be adjacent to the photogate as an electrode. A photogate was placed in the sensing area, and the applied positive voltage bent the potential distribution on the surface. The p-well was set at the ground level and a positive bias was applied to the n-sub to form a potential distribution. The photoelectrons that were generated on the surface side of potential depth *W* were collected on the surface and detected as an electric current from the readout electrode. Although an n-type sensor detects the wavelength intensity by measuring the current on the surface side, the characteristics of the peripheral circuits changed owing to the voltage that was applied to the n-substrate, making it a challenge for us to array the sensor. Additionally, the quantum efficiency may decrease considering that the photocurrent beyond the saddle point depth *W* cannot be detected.

Figure 2a shows a schematic of the proposed filter-free wavelength sensor that was obtained by changing the substrate from an n-type to a p-type silicon substrate in the double-well structure. The sensor proposed a three-layer structure where a deep n-well and p-well were formed on a p-sub silicon substrate to measure the electrons generated by light passing through *W*. A photogate structure was adopted as in the conventional structure, and an *n^+^* diffusion layer was deposited on the deep n-well to detect the electrons generated at a position that was deeper than *W*. Therefore, the peripheral circuit characteristics do not change by providing the n-well in a region that is different from the deep n-well, and the sensor and peripheral circuit can be integrated. Because measuring the light that is passing through the saddle point *W* with a photocurrent is also possible, a quantum efficiency that is higher than that of an n-type silicon substrate sensor can be expected. Furthermore, because the potential depth position can be freely moved while maintaining the potential distribution steeply by the photogate voltage and body biasing, a high sensitivity can also be expected by optimizing the wavelength that is to be detected.

The filter-free wavelength sensor measures the light intensity using the absorption coefficient *α* of silicon according to the wavelength of the light instead of removing the optical filter. The light irradiated on the silicon surface is absorbed inside the silicon to generate electron-hole pairs, and the output photocurrent can be measured. Equation (1) shows the photocurrent *I*_PG_ that is generated based on the depth *W*, which is expressed as:(1)$$I_{\mathrm{PG}}=-\frac{\phi_{0}qS\lambda }{hc}\left(1-e^{-\alpha W}\right)$$

The light intensity *ϕ*_0_ on the silicon surface was calculated by substituting the measured photocurrent.

The photocurrent in the well that passes through the depth *W* is expressed as:(2)$$I_{n-\mathrm{well}}=-\frac{\phi_{0}qS\lambda }{hc}\left(e^{-\alpha W}-e^{-\alpha W_{pn}}\right)$$
where *h* is Planck’s constant, *c* is the speed of light in vacuum, and λ is the light wavelength, *q* is the elementary charge, *W*_pn_ is the junction depth between the p-well and the deep n-well, and *S* is the sensing area.

By calculating the ratio of Equations (1) and (2), we obtain:(3)$$\frac{I_{n-\mathrm{well}}}{I_{\mathrm{PG}}}=\frac{e^{-\alpha W}-e^{-\alpha W_{\mathrm{pn}}}}{1-e^{-\alpha W}}$$

Equation (3) indicates that the ratio of *I*_PG_-to-*I*_n-well_ does not depend on the light intensity, but on the potential depth *W* and silicon absorption coefficient α. In other words, because the current ratio does not depend on the light intensity, it is possible to detect the wavelength by calculating the ratio of the currents *I*_PG_ and *I*_n-well_.

## 3. Fabrication

### 3.1. Device Simulation

A three-dimensional device simulation was conducted using SPECTRA (Link Research, Japan) to evaluate the current characteristics and effectiveness of the sensor with the proposed structure. The light source conditions that were irradiated to the sensing area had a diameter of 100 μm, a light intensity of 1 mW/cm^2^, and a wavelength in the 450–750 nm range. Figure 3a shows the change in the *I*_PG_ and *I*_n-well_ current and its current ratio according to the wavelength. As the wavelength increased, the penetration depth of the light irradiated onto the silicon substrate increased. Therefore, the surface–side current *I*_PG_ decreased and the substrate current *I*_n-well_ increased. The simulation results indicate that the current ratio in the 450–750 nm wavelength range changed from 0.09 to 3.96. Consequently, we were able to identify the wavelengths in the visible light region by calculating the respective current ratios.

### 3.2. Process Simulation and Fabrication

The semiconductor process conditions were determined using the process simulator TCAD (Taurus TSUPREM-4) to fabricate a filter-free wavelength sensor with a double-well structure. To form a double-well structure, the impurity concentration must be in the order of p-well > deep n-well > p-substrate. Because the p-sub silicon that was used was 2.24 × 10^14^ cm^−3^, the impurity concentrations of the deep n-well and p-well were aimed at 10^15^ and 10^16^ cm^−3^, respectively. Table 1 lists the ion implantation conditions of the fabricated sensor according to these requirements. The proposed double-well structure requires a deep n-well junction depth to prevent the p-well and p-sub junctions. The parameters affecting the impurity concentration in the process include the dose amount, the acceleration voltage, the implantation angle as the ion implantation conditions, and the time and temperature as the drive-in conditions. Among them, the parameters that affect the n-well junction depth are the acceleration voltage, time, and temperature. Because the deep n-well aimed to form a depth of 7 µm, the ion implantation was performed with an accelerating voltage of 150 keV and a dose of 1.0 × 10^12^ cm^−2^. The drive-in was performed at 1150 °C for 1530 min. Because the p-well also aimed to form a depth of 2.5 µm, the ion implantation was performed with an accelerating voltage of 80 keV and a dose of 2.0 × 10^12^ cm^−2^. The drive-in was performed at 1150 °C for 270 min. The analysis was performed using secondary ion mass spectrometry (SIMS) to determine the impurity concentration in the fabricated sensor. Figure 3b shows the simulation results of the TCAD and SIMS analyses. The values of phosphorus and boron which were obtained by SIMS analysis were approximately identical to those that were assumed by TCAD, and a double-diffusion well structure was fabricated to be identical to the simulation data.

Figure 4 shows the fabrication process of the sensor. The proposed double-well structure sensor was fabricated using the 1-polysilicon, 2-metal process at the LSI facility in the Toyohashi University of Technology, Japan. The drawing rule used 5 μm, and the wafer used a 4-in p-silicon substrate (P100, 60 Ω/cm, 2.24 × 10^14^ cm^−3^). Figure 5 shows the processed wafer and microscope images of the sensor (300 × 300 µm).

## 4. Experimental Results

### 4.1. Sensor Characteristics

#### 4.1.1. Single Wavelength

To evaluate the fabricated sensor, we measured its wavelength resolution, wavelength detection range, light intensity dependence, and response characteristics. A voltage of 3 V was applied to each output electrode to detect the *I*_PG_ and *I*_n-well_ currents. Furthermore, a PG voltage of 3 V was applied to form a potential distribution of the sensor, and the p-well and p-type silicon substrates were set at the ground level. Each wavelength was irradiated using a laser-driven tunable light source (LDTLS; Tokyo Instruments, Japan). The FWHM of each wavelength range was 5–10 nm. The LED light source was passed through a 400 μm optical fiber (M28L01, Thorlabs, Newton, NJ, USA) with a 20× objective lens (SLMPlan, Olympus, Japan), and the sensing area was irradiated with a 20 μm light source. The current measurements and the control of the sensor were performed using a semiconductor parameter analyzer (B1500A, Keysight, Santa Rosa, CA, USA) The experiments were conducted in a dark room at room temperature.

To measure the wavelength resolution of the sensor, the current ratios of *I*_PG_ and *I*_n-well_ were measured when 0.1 nm increments changed the incident wavelength from 550 nm and 650 nm, as shown in Figure 6a,b, respectively. As the wavelength shifted by 1 nm, the current ratios of 0.0083 and 0.0167 changed at 550 nm and 650 nm, respectively. Additionally, the coefficient of determination was 0.999. Because the change in the current ratio according to the noise of the sensor measurement system occurred at a decimal point of fewer than four digits, the wavelength resolution of the sensor was expected to be 0.1 nm or more.

Figure 6c shows the dependence of the current *I*_PG_ and *I*_n-well_ ratios on the wavelength. The current ratio changed from 0.081 to 8.033, depending on the wavelength from 460 nm to 800 nm respectively. Furthermore, the ratio of *I*_PG_-to-*I*_n-well_ changed depending on the absorption coefficient of the silicon, as described in Equation (3) [29]. In other words, a proportional relationship was confirmed with the light absorption depth (1/*e*), depending on the wavelength. The dependence of the current ratio on the light intensity was evaluated by changing the light intensities to approximately −20 and −40 dB using the neutral density (ND) filters (ndk01, Thorlabs, Newton, NJ, USA). Because the light absorption depth of silicon was constant, the current ratio did not change, even if the light intensity changed at the same wavelength. This indicates that the proposed double-well structure sensor enables the detection of a single wavelength under changing light intensity conditions.

Figure 6d shows the response characteristics of the sensor depending on the light intensity. The *I*_PG_ current occurring from the depth *W* to the surface side was calculated to be 0.05, 0.08, 0.04, and 0.07 A/W at the 490, 530, 590, and 690 nm wavelengths, respectively. This value has the same response characteristics as those of the previously reported single-well-structured sensor [28]. Because the proposed wavelength detection method simultaneously measured the surface–side current *I*_PG_ and the substrate–side current *I*_n-well_, improved current response characteristics can be expected. The response characteristics by the measured current *I*_PG_ and *I*_n-well_ were 0.07, 0.13, 0.17, and 0.31 A/W, and the sensitivity was 1.39, 1.64, 2.5, and 4.18 times higher than that of the previous sensor, respectively. The increased sensitivity had the same value as the current ratio of the sensor according to the wavelength, as shown in Figure 6c. Therefore, it was confirmed that the response characteristics were improved by 0.081–8.033 times when they compared to that of the conventional sensor in the 460–800 nm wavelength range. Each datum was measured ten times at a single wavelength, and the average standard deviation of the current ratio was calculated as 0.00018.

A programmable light source (OSVISX, OneLight Spectra, Vancouvar, BC, Canada) was used to evaluate the current ratio owing to the FWHM change of a single wavelength. The current ratio was measured by irradiating three light sources with central wavelengths of 450, 500, and 550 nm and a light intensity of 15 mW/cm^2^. Figure 7 shows the result of measuring the FWHM of the light source five times every 5 nm from 10 nm to 30 nm. Based on the current ratio at a wavelength of 500 nm, which was used as a reference, when the FWHM increased from 10 nm to 30 nm, the current ratio changed by −0.004, and the error rate was 1.48%. In a previous study on a single-well structure, the error rate was approximately 5.11%, and it was reduced by approximately 3.63% [28]. This result confirmed that the proposed sensor has low dependence on the change in FWHM when it is compared to the single-well structure sensor.

#### 4.1.2. Multiple Wavelength

A general fluorescence detection method measures the intensity of the fluorescence passing through an optical filter by irradiating the detection target with excitation light. Because the proposed sensor does not use an optical filter, it was necessary to simultaneously detect the wavelengths of the excitation light and fluorescence. An LED light source with two wavelengths was irradiated using a sensor and a spectrometer for a comparative analysis to examine the applicability of the fluorescence detection. Figure 8a shows the spectral results of the 490 nm LED light source as the excitation light and the 530 nm or 590 nm LED light source as the fluorescent light as detected by the spectrometer. Because the spectra of the wavelengths of 490 nm and 530 nm are relatively close, as the intensity of the 530 nm wavelength increases, the light intensity of the 490 nm wavelength which was used as the excitation light also increases simultaneously. Because the spectra at wavelengths of 490 nm and 590 nm did not overlap, the spectra were distributed independently. In general, to quantify a spectrum with multiple peaks, the centroid wavelength *λ*_c_ is calculated by a weighted mean of a spectral, as shown in Equation (4) [30]:(4)$$\lambda_{C}=\frac{\left[\int_{\lambda_{1}}^{\lambda_{2}}\lambda \phi \left(\lambda \right)\;d\lambda \right]}{\left[\int_{\lambda_{1}}^{\lambda_{2}}\phi \left(\lambda \right)\;d\lambda \right]}$$

In the case of the localized surface plasmon resonance sensors and nanohole biosensors, the centroid wavelength was adopted and quantified to detect the spectrum of the passing light which was changed by the molecular adsorption with high sensitivity [31,32]. Therefore, the centroid wavelength of the measured spectrum was calculated, and a comparative analysis was performed using the current ratio of the proposed sensor. Figure 8b shows the data comparing the centroid wavelength and the current ratio of the sensor. According to the LED light sources, the centroid wavelength was changed from 497.7 nm to 535.2 nm, and the current ratio with PG 1V was simultaneously applied, and it had changed from 0.446 to 0.863. The proposed method had a high coefficient of determination value of 0.9997. This confirmed the possibility of measuring two different wavelengths by using the current ratio of the proposed sensor, as well as measuring the fluorescence with a relatively wide FWHM wavelength.

In the case of a fluorescence experiment, the intensity of the emitted light (fluorescence) is weaker than the excitation light is. We evaluated the wavelength separation ability of the sensor using two light sources (λ: 490, 530 nm). The wavelength separation ability was calculated by detecting the change in the current ratio according to the wavelength. The intensity of the exciting light (490 nm) was fixed at 7.714 µW, and only the fluorescence light source (530 nm) was reduced from 7.709 µW to 0.0008 µW. The standard deviation average value of the current ratio was calculated to be 0.00015 from the results of each measurement which were taken ten times. Therefore, we considered that the practical measurement value of the current ratio is a measurable ratio of up to four decimal places. Figure 9 shows the current ratio change according to the light intensity. It is considered that the separation ability of 490 nm and 530 nm can be measured up to 1977.95: 1 if up to four decimal places are assumed as significant digits. This result has a higher sensitivity than the 1300:1 measured value of the wavelength separation ability of the presented sensor with an n-type silicon substrate [28].

### 4.2. Fluorescence Reagent

#### 4.2.1. Measurement Configuration

This study detected various fluorescent reagents without optical filters using a single-pixel sensor. The fluorescent reagents Alexa Fluor 488, 594, and 680 (AF488, AF594, and AF680, Thermo Fisher, Waltham, MA, USA) were used to evaluate the fluorescence wavelength detection ability of the sensor. The reagent was maximally excited at wavelengths of 490, 590, and 679 nm, and the emitted fluorescence at the wavelengths of 525, 612, and 702 nm, respectively. Figure 10 shows a schematic of the experimental system which was built to perform a quantitative evaluation. The peak wavelengths of the LED light source (M490F3, M590F3, M680F3, Thorlabs, Newton, NJ, USA) that were used as excitation light sources were 490, 590, and 680 nm (FWHM: 26, 16, and 22 nm), and the light intensities were 21.28, 15.05, and 13.60 µW/cm^2^, respectively. The reagent that was to be analyzed was 2.5 mL of deionized water (DIW) and seven types of fluorescent reagents (10, 5, 2, 1, 0.5, 0.2, and 0.1 μM) in a standard quartz cell (T-5-UV-10, TOSOH, Japan). The excitation light and fluorescence passing through the quartz cell were irradiated onto the sensor and spectrometer (OCEAN HDX, Ocean Photonics, Tokyo, Japan) using a two-branch optical fiber (BIF600-UV/VIS, Ocean Photonics, Japan), and the current and spectral characteristics of the sensor were measured simultaneously. The fluorescent reagent measurements were conducted in a dark room at room temperature.

#### 4.2.2. Measurement Results

A typical fluorescence sensor detected the light intensity of the fluorescence that passes through the optical filters to quantify and identify the target. However, the proposed sensor quantified the fluorescence by detecting the current ratio according to wavelength change when the excitation light and fluorescence were simultaneously irradiated. Therefore, before we evaluated the concentration dependence of the fluorescence reagent, we experimented with the following three assumptions: (i) When the optimal excitation wavelength is irradiated to the fluorescent reagent, the higher the concentration of the fluorescent reagent is, then the more the excitation light is sufficiently absorbed, and a strong fluorescence is emitted. It was assumed that the current ratio of the sensor increased as the concentration increased; (ii) when a fluorescent reagent is irradiated with a shorter wavelength than the optimized excitation wavelength, the absorption rate of the excitation light increases as the concentration of the fluorescent reagent increases. However, the current ratio decreases as the concentration decreases because the absorption rate on the long wavelength side of the irradiated excitation light spectrum is high, and the fluorescence intensity is weak; (iii) when a fluorescent reagent is irradiated with excitation light with a wavelength that is longer than the fluorescence wavelength, most of the excitation wavelengths pass through the fluorescent reagent. Therefore, the fluorescence is not emitted, and the excitation light is transmitted. Even if the concentration of the reagent is high, the current ratio is not expected to change, considering that most of the excitation light passes through the fluorescent reagent.

Figure 11 shows the current ratio of the sensor by the irradiation of each LED light source (λ = 490, 590, and 680 nm) to each fluorescent reagent (AF488, AF594, and AF680), and the simultaneously measured spectra are shown in Figure 12. The result of irradiating each LED light source to the DIW was used as a reference (red dotted line), and the fluorescence reagent was identified and quantified using the current ratio for each concentration. The detection limit of the sensor is considered to be above 0.1 μM because the measured current ratio varies linearly up to 0.1 μM concentration for all of the reagents.

As shown in Figure 11a, the current ratio increased from 0.31 to 0.36 by reagent concentration being from 0.1 to 10 μM. Because AF488 was irradiated with a light source at 490 nm, the optimal excitation wavelength, most of the excitation light was absorbed by the reagent, and long-wavelength fluorescence was emitted. Because the normal distribution of the spectrum passing through the reagent shifts to a longer wavelength, the current ratio increases. Additionally, the spectrum was shifted to a longer wavelength as the reagent concentration changed from 0.1 to 10 μM, as shown in Figure 12a. Therefore, it is possible to identify and quantify a fluorescent reagent by calculating the current ratio at the optimal excitation wavelength. Figure 11e,i shows a similar result as that in Figure 11a for the optimized excitation wavelength, which is related to assumption (i). The current ratio increased from 1.39 to 1.42 in AF594 and 3.22 to 3.45 in AF680. Moreover, the spectral characteristics in Figure 12e,i have confirmed that the normal distribution shifted toward a longer wavelength side rather than an excitation wavelength.

Figure 11b shows that the current ratio was consistent with the current ratio of the DIW owing to the concentration change. Because a wavelength that was longer than the optimal excitation wavelength was irradiated, most of the excitation light was passed to the reagent. The slight change in the current ratio at high concentrations (2–10 μM) was caused by the absorption of the short wavelength side in the excitation light. Furthermore, the absorption on the short wavelength side was confirmed in the result, as shown in Figure 12b. As shown in the result of Figure 11c,d, the change in the current ratio was insignificant since most of the excitation wavelengths passed through the reagent, as shown in Figure 11b, and its relevance to assumption (iii) can be confirmed.

Figure 11d shows the current ratio of the 470 nm LED irradiation with a wavelength that was shorter than the optimal excitation wavelength of AF594. In the spectrum of the irradiated 470 nm LED excitation light, a relatively large amount of the wavelength on the long wavelength side was absorbed, and hence, the decrease in the current ratio was confirmed as the concentration of the reagent increased. Furthermore, the absorption on the long wavelength side was confirmed in the spectral results in Figure 12d. As shown in the result of Figure 11g,h, the long wavelength component was absorbed by the short excitation wavelength, and its relationship with assumption (ii) can be confirmed in Figure 11d.

Figure 13 shows the relationship between the centroid wavelength (Figure 12) and the current ratio (Figure 11) depending on the fluorescent reagent concentration. In Section 4.1.2, the centroid wavelength of the spectral curve was proportional to the sensor ratio for a wavelength with a relatively wide FWHM. Similarly, because the normal distribution of the spectral curve shifts according to the change in the concentration of the fluorescent reagent, the current ratio which was measured by the sensor was proportional to the centroid wavelength. The measurement results can be divided into three primary patterns using the LEDs with three different excitation wavelengths. Therefore, we showed that the current ratio of the proposed sensor can be used to detect the fluorescent reagent and its concentration.

## 5. Conclusions

In this study, we proposed a filter-free wavelength sensor with a double-diffusion well structure and evaluated a new wavelength detection method. The proposed structure was simulated using SPECTRA, which showed the possibility to achieve wavelength identification in the proposed structure. To fabricate the sensor, TCAD was used to optimize the impurity concentration and design the fabrication process. The impurity concentration of the sensor which was fabricated using the SIMS analysis results was consistent with the simulation results. The proposed double-well structure sensor was fabricated using the 1-polysilicon, 2-metal process at the LSI facility of Toyohashi University of Technology, Japan. The current ratio was obtained according to the wavelength by measuring the *I*_PG_ and *I*_n-well_ of the manufactured sensor. The result confirmed that this ratio depended on the absorption depth of the silicon at each wavelength, and it did not depend on the light intensity. The low FWHM dependence of the wavelength in the proposed structure confirmed the possibility to achieve fluorescence detection with a broad wavelength. A spectrum with two peak wavelengths was calculated as the centroid wavelength, and it was compared with the current ratio of the sensor, which showed high linearity. Therefore, it is possible to quantify the wavelength with a relatively wide FWHM using the proposed sensor. As an application experiment, a quantitative evaluation was performed using three types of fluorescent reagents. The fluorescent reagents were irradiated with three excitations of three types of LED light to evaluate the reagent concentration dependence and the spectral properties, simultaneously. Furthermore, the current ratio of the sensor was detected by the excitation light and the fluorescence emitted from the reagent, and they were compared with the spectral characteristics. Additionally, the ratio change was 0.31 to 0.36 in AF488, 1.39 to 1.42 in AF594, and 3.22 to 3.45 in AF680, respectively, depending on the concentration of the reagent. This indicated that the concentration of the reagent by the fluorescence can be detected from the current ratio, thereby suggesting that various fluorescence signals can be detected. The proposed sensor can be applied in biofields such as POCT as a miniaturized wavelength detection sensor that does not use optical components. In the future, the development of a miniaturized optical detection system that is capable of imaging wavelength information by arranging the proposed single-pixel structure sensor is expected to be conducted.

## Figures and Tables

**Figure 1 biosensors-12-01033-f001:**
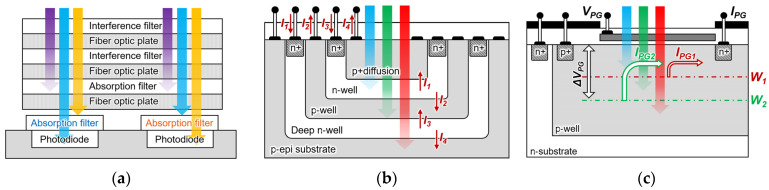
Principle schematic of the on-chip fluorescence detection device. (**a**) Description of what is contained in the first panel; (**b**) CMOS Buried quad p-n Junction photodetector; (**c**) filter-free fluorescence sensor.

**Figure 2 biosensors-12-01033-f002:**
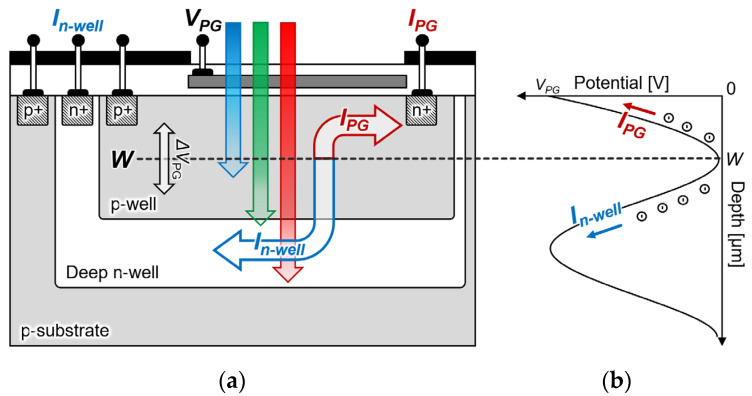
Proposed filter-free wavelength sensor: (**a**) cross-sectional schematic; (**b**) potential distribution.

**Figure 3 biosensors-12-01033-f003:**
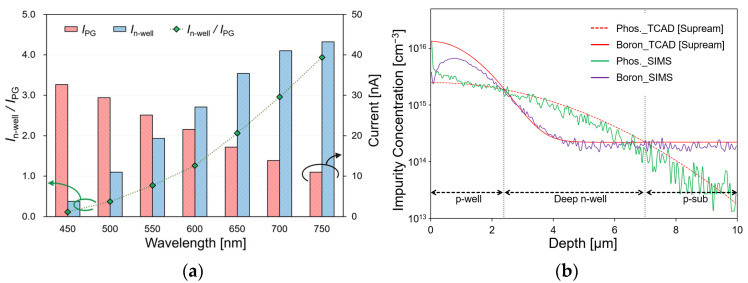
(**a**) Simulation results of the current characteristics of the sensor; (**b**) simulation results by TCAD and the analysis results by SIMS of the impurity concentration of the fabricated sensor.

**Figure 4 biosensors-12-01033-f004:**
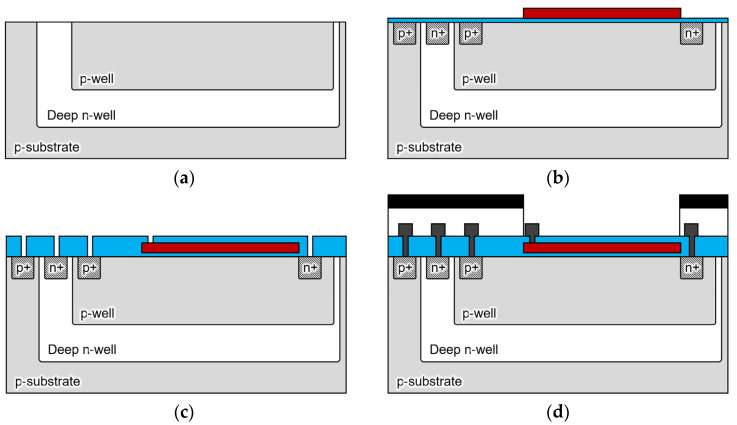
Cross-sectional schematic of the sensor fabrication process. (**a**) A p-type 4-in silicon wafer (60 Ω/cm) is ion-implanted and driven in under the conditions that are shown in Table 1 to form a double-diffusion well structure; (**b**) the gate oxide film of 60 nm and Poly-Si of 350 nm are deposited, and the gate electrode is formed by phosphorous diffusion; (**c**) 400-nm-thick Tetra ethoxy silane (TEOS) is deposited by low-pressure chemical vapor deposition (LP-CVD), and contact holes are opened by oxide film etching; (**d**) after Al wiring, a light shielding film is formed by Al sputtering of 1.0 µm outside the sensing area.

**Figure 5 biosensors-12-01033-f005:**
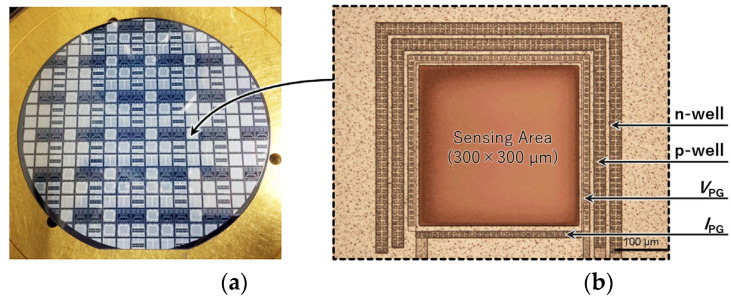
Fabricated filter-free wavelength sensor. (**a**) A picture of the 4-in wafer before the chip dicing process; (**b**) microscope image of the sensing area.

**Figure 6 biosensors-12-01033-f006:**
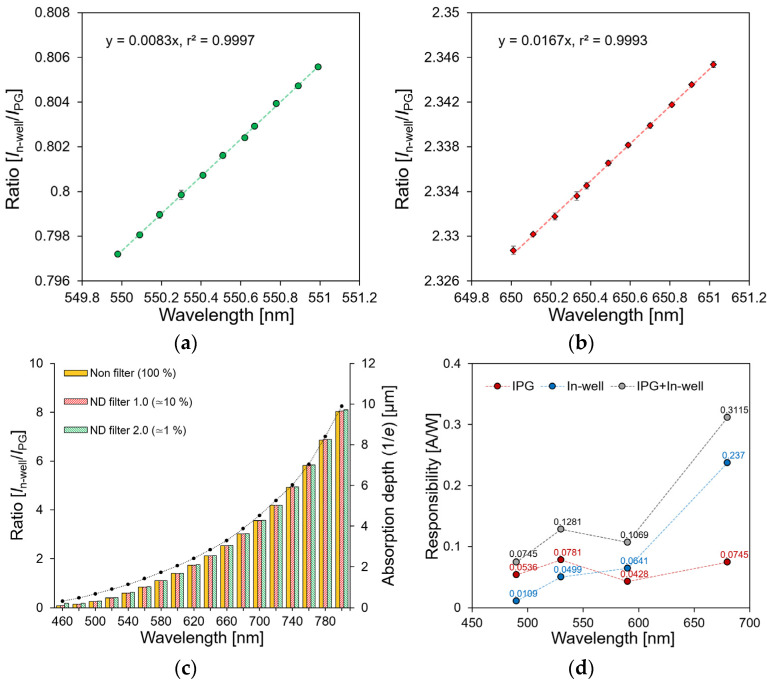
Wavelength dependence of current ratio at: (**a**) 550 nm; (**b**) 650 nm; (**c**) 460~800 nm. (**d**) Response characteristics of the sensor.

**Figure 7 biosensors-12-01033-f007:**
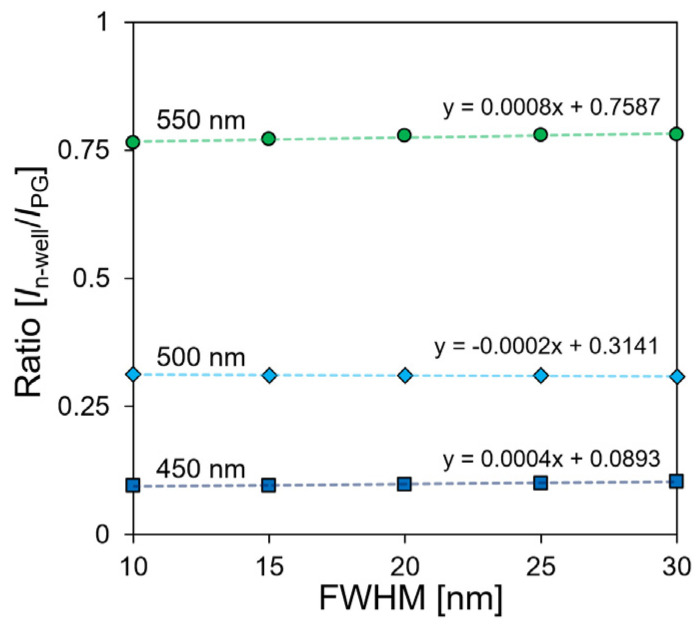
FWHM dependence of current ratio.

**Figure 8 biosensors-12-01033-f008:**
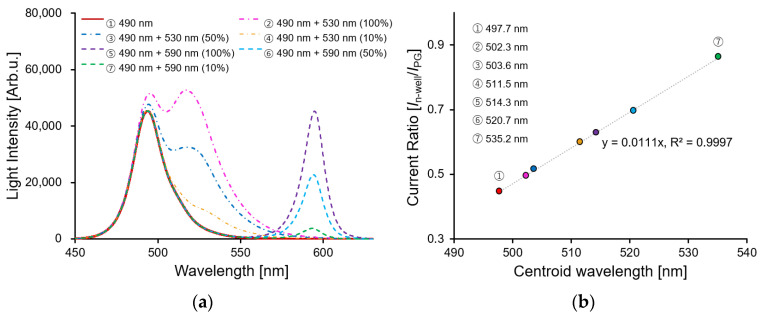
(**a**) Spectral characteristics by LED light source 490 nm, 490 + 530 nm, and 490 + 590 nm. The 490, 530, and 590 nm light intensities are 16.98, 13.44, and 22.66 µW/cm^2^, respectively. The 530, and 590 nm LED light sources changed the light intensity by 100, 50, and 10%; (**b**) the relationship between centroid wavelength by spectrum and current ratio.

**Figure 9 biosensors-12-01033-f009:**
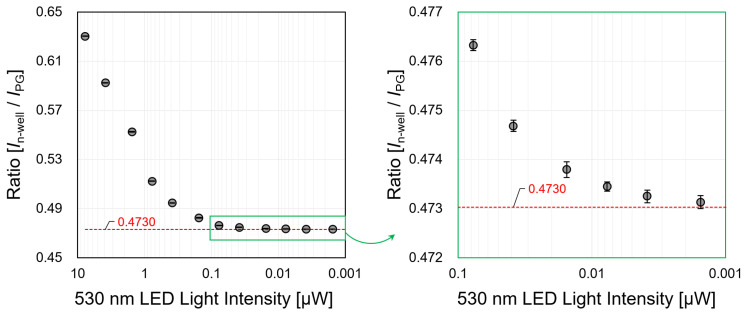
Change in current ratio when the LED light source is irradiated with wavelengths of 490 nm (7.714 µW) and 530 nm (0.0008–7.7090 µW), simultaneously.

**Figure 10 biosensors-12-01033-f010:**
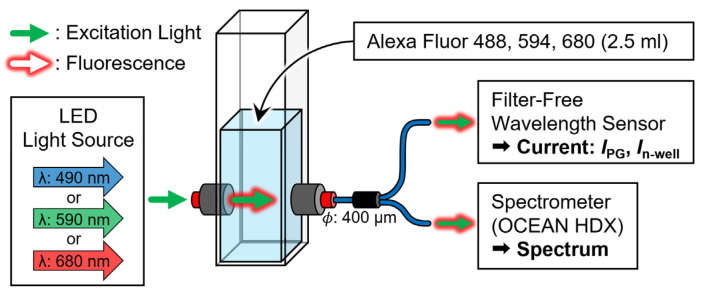
Schematic of an evaluation system to compare the current ratio of the sensor and spectral characteristics of the spectrometer for light emitted from a fluorescent reagent.

**Figure 11 biosensors-12-01033-f011:**
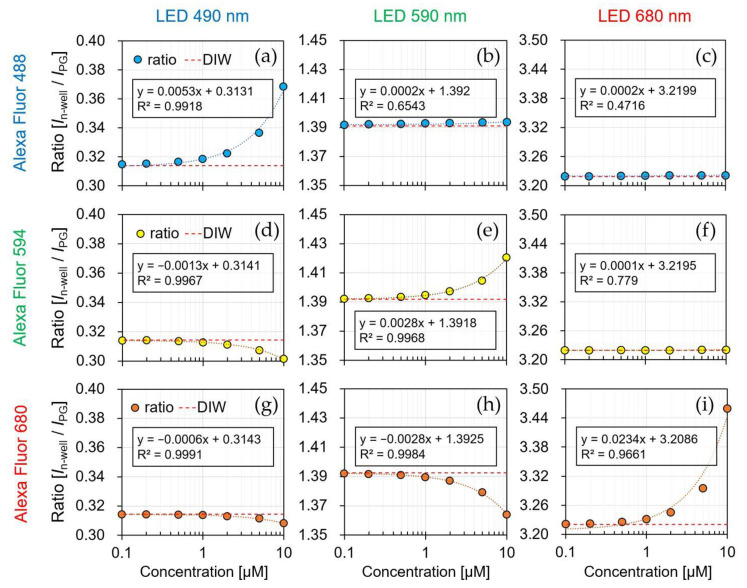
Current ratio by the concentration of fluorescent reagent. (**a**) AF488-*λ*_em_: 490 nm; (**b**) AF488-*λ*_em_: 590 nm; (**c**) AF488-*λ*_em_: 680 nm; (**d**) AF594-*λ*_em_: 490 nm; (**e**) AF594-*λ*_em_: 590 nm; (**f**) AF594-*λ*_em_: 680 nm; (**g**) AF680-*λ*_em_: 490 nm; (**h**) AF680-*λ*_em_: 590 nm; (**i**) AF680-*λ*_em_: 680 nm.

**Figure 12 biosensors-12-01033-f012:**
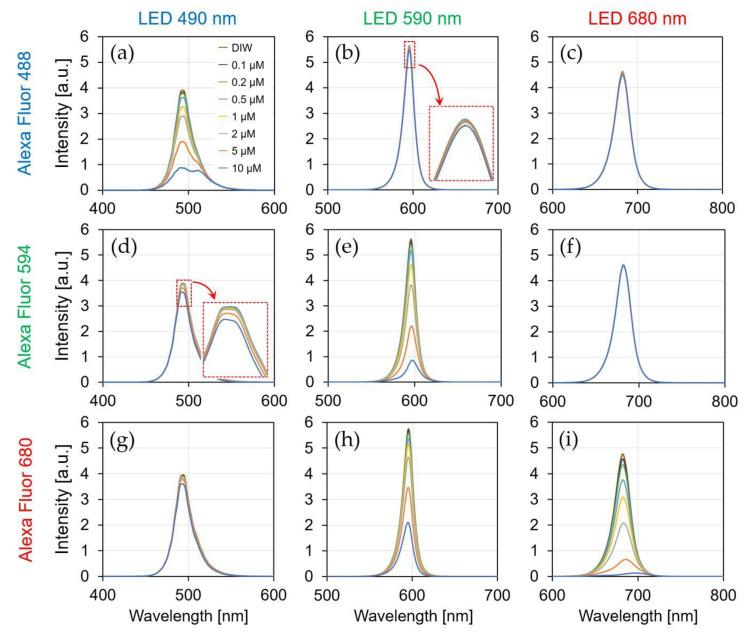
Spectral characteristic by the concentration of fluorescent reagent. (**a**) AF488-*λ*_em_: 490 nm; (**b**) AF488-*λ*_em_: 590 nm; (**c**) AF488-*λ*_em_: 680 nm; (**d**) AF594-*λ*_em_: 490 nm; (**e**) AF594-*λ*_em_: 590 nm; (**f**) AF594-*λ*_em_: 680 nm; (**g**) AF680-*λ*_em_: 490 nm; (**h**) AF680-*λ*_em_: 590 nm; (**i**) AF680-*λ*_em_: 680 nm.

**Figure 13 biosensors-12-01033-f013:**
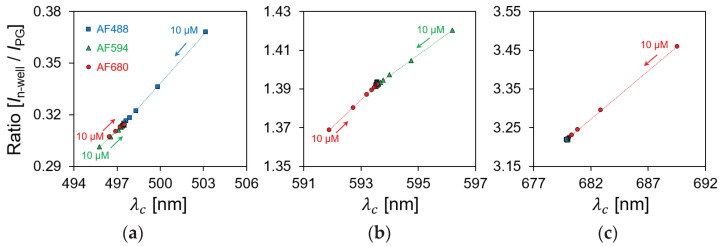
Relationship between centroid wavelength and current ratio depending on fluorescent reagent concentration. (**a**) LED light source of 490 nm; (**b**) LED light source of 590 nm; (**c**) LED light source of 680 nm.

**Table 1 biosensors-12-01033-t001:** Ion implantation conditions.

	Deep n-Well	p-Well
Types of impurities	Phosphorus	Boron
Dose [cm^−2^]	1.0 × 10^12^	2.0 × 10^12^
Acceleration voltage [keV]	150	80
Injection angle [deg]	7	7
Drive-in time [min]	1530	270
Drive-in temperature [°C]	1150	1150

## Data Availability

Not applicable.

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
