# Peer review of "Demonstrating a Filter-Free Wavelength Sensor with Double-Well Structure and Its Application"

_biosensors, 2022, doi:10.3390/bios12111033_

Round 1

Reviewer 1 Report

In this manuscript, Sawada et al. report an improved filter-free wavelength sensor with a double-well structure for detecting fluorescence without an optical filter. Recently, an on-chip detection system based on a CMOS image sensor is attracted much attention because of its point-of-care testing (POCT) application, where fluorescence is detected directly by the sensor and thus no bulky optics are required, enabling system miniaturization. This manuscript is interesting, and the experimental work is thorough. In my view, the science presented in this manuscript is sufficiently novel to warrant publication in Biosensors. However, in view of this referee, the technical quality of the paper has to be improved prior to publication. In the following, I will mention just a few of the numerous flaws and inconsistencies I noted:

(1)  The following sentence appeared two times in lines 132 and 137; “The light intensity Ï•0 on the silicon surface was calculated by substituting the measured 132 photocurrent”. Please remove one sentence.

(2)  Page 11, line 378, please remove the “=” sign from the sentence.

(3)  What is the separation ability of excitation light and fluorescence for this sensor?

(4)  What is the detection limit of this sensor to low concentrations of the fluorescent reagents? Is this filter-free wavelength sensor suitable for detecting the emission of less emissive dyes?

(5)  Page 8, line 261-262, the authors claim that ‘we can conclude that the current ratio does not change even if FWHM is widened, centering on the peak wavelength.’ Is there any rationale, or is it just an observation/experimental fact? As we know, if the width of FWHM increases, a reduction in sensitivity and errors in the calculation are expected.

(6)  At which conditions the measurement setup was performed?

(7)  The references are not uniform and properly cited. 

Reviewer 2 Report

The authors demonstrated a filter-free wavelength sensor with a double-diffusion well structure and evaluated a new wavelength detection method. This is an interesting study considering that the proposed sensor could have the potential to be widely used as POCT for chemical and biosignal detection with simplified structures. Overall, the study is well structured, but several comments should be addressed:

1.     What is the operating temperature range for this type of sensor? Have the authors conducted the calibration of sensors at different temperatures? Is there any temperature dependence?

2.     What about the dynamic range of the photo detective sensor?

3.     The doping profile and depth should be critical in this type of sensor. Could the authors comment on the difficulty in fabricating the double well structure? Will the ion implantation be uniform and controllable over a full wafer?

4.     Please also add the measurement error bar for each set of data points in Figures 6, 7, 8 , 10 if there are multiple measurements.

5.     Could the authors explain a little bit about the figure captions in Figure 8a?

6.     There is a typo in Figure 1 ( Figure 1 a,b,c instead of Figure 1 a, b, a)

Reviewer 3 Report

The results of this manuscript show that the proposed method can detect single wavelengths in the 460–800 nm range. Actually, in this range of optical detection, many photodetectors have been reported, such as silicon-based tunneling device (DOI: 10.1109/LED.2022.3203474), quantum dot phototransistors (Adv. Optical Mater. 2018, 6, 1800985) and the hybrid photovoltage triodes (Nat. Commun. 2021, 12(1), 6696). I suggest the authors make comments in the introduction portion or at least complement the references of the related devices.

Reviewer 4 Report

Y-J Choi et. al design a novel double well sensor whose current ratios are dependent on wavelength and independent of intensity. They thoroughly characterize the design and demonstrate its behavior to be as expected from simulations and theoretical calculations. I have one major concern about the applicability of such a device as discussed below. Other than that, the design is novel and well-studied.

In order for this device to replace a filter, we would need to solve the inverse problem of estimating the wavelength given the measured current ratios. For a monochromatic source, this seems to work quite well. However, often the source is not monochromatic, or in the case of fluorescence experiments, the emitted light is much weaker than the excitation light. In such scenarios, it seems like the developed device wouldn't be able to pick out the light at a specific wavelength. 

Round 2

Reviewer 4 Report

Dear authors,

Thank you for addressing my concerns and especially demonstrating that you have very good sensitivity when you have two wavelengths with a dynamic range of over a 1000. I think it is a neat result!